# Osteosarcoma-Derived Extracellular Vesicles Induce Lung Fibroblast Reprogramming

**DOI:** 10.3390/ijms21155451

**Published:** 2020-07-30

**Authors:** Alekhya Mazumdar, Joaquin Urdinez, Aleksandar Boro, Jessica Migliavacca, Matthias J.E. Arlt, Roman Muff, Bruno Fuchs, Jess Gerrit Snedeker, Ana Gvozdenovic

**Affiliations:** 1Department of Orthopedics, Balgrist University Hospital, CH-8008 Zurich, Switzerland; Alekhya.Mazumdar@balgrist.ch (A.M.); jurdinez.unq@gmail.com (J.U.); alexandarboro@yahoo.com (A.B.); MArlt@gmx.net (M.J.E.A.); Roman.Muff@balgrist.ch (R.M.); fuchs@sarcoma.surgery (B.F.); Jess.Snedeker@balgrist.ch (J.G.S.); 2Institute for Biomechanics, ETH Zurich, CH-8008 Zurich, Switzerland; 3Department of Oncology, Children’s Research Center, University Children’s Hospital Zürich, CH-8032 Zurich, Switzerland; Jessica.Migliavacca@kispi.uzh.ch

**Keywords:** osteosarcoma, extracellular vesicles, lung fibroblasts, tumor microenvironment, tumor-host interactions

## Abstract

Tumor-secreted extracellular vesicles (EVs) have been identified as mediators of cancer–host intercellular communication and shown to support pre-metastatic niche formation by modulating stromal cells at future metastatic sites. While osteosarcoma, the most common primary malignant bone tumor in children and adolescents, has a high propensity for pulmonary metastases, the interaction of osteosarcoma cells with resident lung cells remains poorly understood. Here, we deliver foundational in vitro evidence that osteosarcoma cell-derived EVs drive myofibroblast/cancer-associated fibroblast differentiation. Human lung fibroblasts displayed increased invasive competence, in addition to increased α-smooth muscle actin expression and fibronectin production upon EV treatment. Furthermore, we demonstrate, through the use of transforming growth factor beta receptor 1 (TGFBR1) inhibitors and CRISPR-Cas9-mediated knockouts, that TGFβ1 present in osteosarcoma cell-derived EVs is responsible for lung fibroblast differentiation. Overall, our study highlights osteosarcoma-derived EVs as novel regulators of lung fibroblast activation and provides mechanistic insight into how osteosarcoma cells can modulate distant cells to potentially support metastatic progression.

## 1. Introduction

Osteosarcoma is an aggressive primary bone malignancy mainly affecting children and adolescents [1]. Pulmonary metastases, the major cause of death in osteosarcoma patients, are detectable already in 20% of patients at diagnosis and the majority of patients with localized disease develop metastases later on. The prognosis for patients with metastatic disease remains poor despite the enormous clinical efforts to improve treatment efficacies by varying combinations and dosages of established chemotherapeutics [2]. The identification of molecular therapeutic targets has been challenging due to the significant degree of tumor heterogeneity, genomic instability and a high frequency of chromothripsis in osteosarcoma cells. Indeed, only a few recurrent clinically actionable alterations are present, and clinical trials with targeted agents so far have been generally unsatisfactory [3]. It is now well established that the interplay between the tumor and stromal cells within the local and distant tumor microenvironment plays a pivotal role in cancer progression and metastasis [4]. Unlike tumor cells, the non-cancer stromal cells within the tumor are genetically stable, and, therefore, targeting these normal cells and their interactions with tumor cells might offer a more efficient strategy to treat metastatic disease. 

Accumulating evidence indicates that cancer–host intercellular communication is mediated by tumor-secreted extracellular vesicles (EVs), including exosomes [5]. Exosomes are small membrane EVs of multivesicular endosomal origin with a size range of 40–150 nm that carry biomolecules such as proteins, nucleic acids, lipids and metabolites [6,7]. Upon cellular uptake, the exosomal cargo is transferred to the recipient cells and can trigger cell phenotypic changes. Exosomes can also modulate nearby or distant cells by direct contact of their surface molecules and thereby activate intracellular signaling pathways in target cells. Numerous studies employing various cancer models have highlighted cancer-derived exosomes as mediators of metastasis [8]. By reprogramming resident or recruited stromal recipient cells, EVs support pre-metastatic niche (PMN) formation at future metastatic sites, which provides a favorable microenvironment that subsequently promotes the survival and outgrowth of disseminated tumor cells [9,10,11].

In recent years, the functional relevance of tumor-derived exosomes for osteosarcoma progression has drawn increasing attention within the field [12]. Because osteosarcoma initiation and development are associated with dysregulated bone remodeling, several studies have experimentally addressed the effects of osteosarcoma cell-derived EVs on neighboring cells in the local bone tumor microenvironment. EVs isolated from osteosarcoma cells were shown to contain a pro-osteoclastogenic cargo or specific miRNAs that could promote a osteoclast gene expression profile in murine macrophages, induce osteoclast formation and stimulate their bone resorption activity [13,14]. The metastatic osteosarcoma-derived exosomes, but not non-metastatic osteosarcoma-derived exosomes, significantly increased the migration and invasion of human osteoblast cells in vitro [15]. In addition to an effect on bone remodeling, EVs can modulate tumor angiogenesis. Namely, EV-mediated cross-talk between osteosarcoma and endothelial cells stimulated the release of pro-angiogenic factors and potentiated tube formation of endothelial cells [14]. Interestingly, osteosarcoma-derived EVs trigger a pro-metastatic inflammatory loop by altering the physiology of mesenchymal stem cells [16]. Osteosarcoma-derived EVs have also been reported to induce a tumor-supporting M2 phenotype in alveolar macrophages, which assists immune escape in tumors [17]. In contrast to the crosstalk between the primary tumor and the bone microenvironment, the EV-facilitated interplay between osteosarcoma cells and cellular components in the lung metastatic microenvironment has been largely unexplored. It is, however, particularly important to understand the osteosarcoma-driven dynamic stromal changes in remote tissue, such as endothelial and fibroblast activation, as these might be crucial for metastasis formation. 

We, therefore, aimed to delineate the role of osteosarcoma cell-derived EVs in intercellular communication between tumor and distant stromal cells. Using robust in vitro approaches, we investigated the potential involvement of EVs in endothelial and lung fibroblast reprogramming as well as the underlying molecular mechanism.

## 2. Results

### 2.1. Osteosarcoma Cell-Derived Vesicles are Internalized by Endothelial Cells and Lung Fibroblasts In Vitro

We first characterized the EVs isolated from the conditioned medium of human 143-B, SaOS-2, HuO9, and murine Dunn and LM8 osteosarcoma cells. The size of vesicles, assessed by nanoparticle tracking analysis (NTA), ranged between 30 and 300 nm, with the peak being around 100 nm (Figure 1A and Appendix A). Among the human cell lines, the highly metastatic 143-B cells have the shortest doubling time and produced a high amount of EVs and were, therefore, chosen for further experiments (Appendix A). The negatively stained transmission electron microscopy confirmed the size of vesicles and revealed their round morphology associated with a lipid bilayer structure, typical for exosomal membrane vesicles (Figure 1B). Moreover, the presence of EV-associated markers such as CD63 and CD81 was detected through flow cytometry analysis of EV-coated aldehyde-sulphate beads (Figure 1C). 143-B EVs were subjected to iodixanol density gradient centrifugation, and the expression of additional EV markers CD9 and ALIX was detected in the gradient fractions corresponding to densities between 1.14 and 1.20 g/mL by Western blotting (Figure 1D). Correspondingly, calreticulin, an endoplasmic reticulum-associated protein, was not found in these fractions, but was detected in the 143-B protein cell lysate. 

To investigate the interaction of osteosarcoma cell-derived EVs with endothelial cells and lung fibroblasts, we performed flow cytometry analysis to measure the internalization of fluorescently labelled 143-B EVs by human umbilical vein endothelial cells (HUVEC), WI-38 and MRC-5 cells over time in vitro. Uptake efficiencies were high in all cells with more than 95% of cells being positive after 24 h, however, HUVEC cells internalized EVs more rapidly compared to lung fibroblasts (Figure 1E–G). Furthermore, we observed both a dose-dependent as well as a time-dependent uptake of EVs by recipient cells. EV internalization was abolished when the cells were either pretreated with Dyngo-4a, an inhibitor of endocytosis, or incubated with EVs at 4 °C, indicating that it is EV uptake that is mediated by an active, energy-dependent endocytic process (Appendix A). 

Together, these results indicate that EVs isolated from 143-B cells were of a high purity and were consistent with the size, structure and molecular characteristics of exosomes, as described in the literature. Moreover, osteosarcoma cell-derived EVs are efficiently taken up by stromal cells.

### 2.2. Osteosarcoma Cell-Derived Vesicles Promote the Functional Activation of Lung Fibroblasts but Not Endothelial Cells In Vitro

Because cell–cell interaction between cancer cells and the activated endothelium in metastasis-prone sites facilitates the metastatic spread [18], we next investigated the potential effects of osteosarcoma cell-derived EVs on the activation of endothelial cells. We analyzed by real-time quantitative reverse transcription polymerase chain reaction (qRT-PCR) the expression pattern of cellular adhesion molecules in HUVEC cells upon incubation with 143-B EVs. Among all endothelial activation markers, only *ICAM1* levels were modestly increased after treatment with 143-B EVs (Appendix A). We then employed a tumor-endothelial cell adhesion assay and, consistently with the above-mentioned findings, pre-incubation of HUVEC cells with EVs did not significantly alter the adhesion of 143-B cells to HUVEC monolayers both in the absence or presence of tumor necrosis factor α (TNFα), a robust endothelial activation factor (Figure 2A). 

We next explored if 143-B EVs affect the functionality of lung fibroblasts using a 3D-spheroid invasion assay. EV incubation lead to a significant increase in the average distance travelled by cells from the center of the spheroid as well as an increase of the total number of cells that were invading (Figure 2B,C). This was consistent for both WI-38 and MRC-5 fibroblasts. In line with these findings, zymography analysis demonstrated increased MMP2 and MMP9 metalloproteinase activity in fibroblasts treated with 143-B EVs compared to untreated control cells (Figure 2D).

In summary, 143-B EVs activate and induce significant phenotypic changes in lung fibroblasts but not in endothelial cells.

### 2.3. Osteosarcoma Cell-Derived Vesicles Cause Differentiation of Lung Fibroblasts into Myofibroblasts/Cancer-Associated Fibroblasts

The EV-mediated increase in the invasive capabilities of normal lung fibroblasts suggested that EVs are able to induce fibroblast differentiation into myofibroblasts or cancer-associated fibroblasts (CAFs). We, therefore, investigated by qRT-PCR EV-driven transcriptional reprogramming of WI-38 and MRC-5 fibroblasts by analyzing the expression of known fibroblast activation genes, as well as fibrosis-related and pro-inflammatory genes commonly associated with CAFs (Figure 3A). Consistent with the myofibroblast/CAF phenotype, we observed an increase in mRNA levels of *ACTA2* (gene encoding α-smooth muscle actin (αSMA)) and fibronectin-1 (*FN1*) in WI-38 and MRC-5 cells upon 143-B EV treatment. We subsequently confirmed 143-B EV-induced elevated αSMA and FN1 protein levels in WI-38 and MRC-5 fibroblasts through Western blotting (Figure 3B). Furthermore, we observed increased protein expression of fibroblast activation protein (FAP), an additional CAF marker (Appendix A). Transforming growth factor β (TGFβ), a known modulator of myofibroblast differentiation, was used as a positive control for fibroblast activation. To further ensure that this effect was not cell line specific, we isolated EVs from a panel of human and murine osteosarcoma cell lines and studied their effects on lung fibroblasts. EVs isolated from poorly metastatic SaOS-2 and Hu09 osteosarcoma cells did not increase αSMA expression in lung fibroblasts (Appendix A). On the contrary, murine Dunn- and LM8-derived EVs efficiently induced fibroblast differentiation (Appendix A). Interestingly, EVs secreted by LM8 cells, a highly metastatic derivative of Dunn cells, had a more profound effect on αSMA expression than Dunn EVs. Moreover, incubation of lung fibroblasts with Dunn and LM8 EVs resulted in a significant increase in the dissemination of fibroblasts using the spheroid invasion assay compared to the untreated control (Appendix A).

Collectively, the data demonstrate that highly metastatic osteosarcoma cells communicate with lung fibroblasts through EVs and regulate fibroblast activation and differentiation into myofibroblasts.

### 2.4. EV-Associated TGFβ Signaling Causes Myofibroblast Differentiation

Since TGFβ has been previously demonstrated to be actively carried by prostate cancer EVs and involved in myofibroblast differentiation [19] and TGFβ has been detected in osteosarcoma EVs [16], we sought to explore whether 143-B EV effects on lung fibroblasts were TGFβ-dependent by using a selective inhibitor of ALK5/TGFβ type I receptor (SB-431542). Inhibition of TGFβ signaling abrogated 143-B-, Dunn- and LM8 EV-mediated αSMA induction in both WI-38 and MRC-5 fibroblasts (Figure 4A, Appendix A). We next examined TGFβ downstream signaling in fibroblasts upon 143-B EV exposure. Indeed, 143-B EVs caused a sharp increase in SMAD2 phosphorylation after 1–2 h in WI-38 and MRC-5 fibroblasts and levels returned to baseline after 24 h (Figure 4B). Additionally, pre-treatment with the SB-431542 inhibitor prevented 143-EV-induced SMAD2 phosphorylation. 

Taken together, these findings demonstrate that EV-associated TGFβ triggers SMAD2-dependent intracellular signaling and directs myofibroblast differentiation in vitro.

### 2.5. CRISPR-Cas9-Mediated Deletion of TGFB1 Prevents Osteosarcoma-Derived EVs from Inducing Myofibroblast Differentiation

To further identify which specific *TGFβ* isoform is responsible for the fibroblast-activating effect of 143-B EVs, we first investigated the expression of respective isoforms at the mRNA level. *TGFB2* and *TGFB3* were poorly expressed and *TGFB1* was found to be the highest expressed *TGFβ* isoform (Figure 5A). Hence, we next carried out the CRISPR-Cas9-mediated gene deletion of *TGFB1* in 143-B cells. Knockout efficiency in whole cell extracts and the respective EVs were confirmed by Western blot (Figure 5B). In contrast to EVs isolated from non-targeting control cells (NT), EVs isolated from TGFB1-knockout cells (KO#1, KO#2) failed to induce αSMA expression in lung fibroblast (Figure 5C). We next investigated TGFβ1 levels in EVs derived from human osteosarcoma cell lines with different metastatic potential by ELISA. Our data indicate that 143-B cell-derived EVs contain higher TGFβ1 levels in comparison to EVs isolated from poorly metastatic SaOS-2 and Hu09 cells (Appendix A).

These findings indicate that osteosarcoma EV-associated TGFβ1 is required for driving fibroblast to myofibroblast differentiation.

## 3. Discussion

The premetastatic niche (PMN) establishment is associated with stromal changes, including the perturbation of endothelial cell function, the reprogramming of stromal cells, extracellular matrix (ECM) remodeling and the recruitment of several cell populations to secondary organ sites [9]. Pulmonary fibroblasts represent a large population of the pre-metastatic stroma and their activation into myofibroblasts/CAFs is one of the early events occurring during PMN formation [20]. Breast cancer cell EVs have been shown to exhibit organotropism and are preferably taken up by resident lung tissue fibroblasts compared to other stromal cells [21,22]. Osteosarcoma cell-derived EVs have been shown to be selectively retained in the lungs of mice after intravenous injections [23]; however, their interaction with fibroblasts from the lung microenvironment remains largely unexplored. Addressing the effects of EVs on stromal cells in an in vivo setting remains, so far, technically challenging, as physiological dosing, administration route, biodistribution as well as the pharmacokinetic clearance of EVs are not well established in the field of osteosarcoma-derived EVs. Additionally, due to the multicellular and molecular complexity of the microenvironment in vivo, it is difficult to acquire explicit evidence of the EV-driven reprogramming of stromal cells and distinguish between direct and indirect EV effects. Despite the inherent limitations of an only in vitro approach, our experimental strategy reduced the complexity and included carefully controlled conditions, which enabled us to explore whether and how EVs affect stromal cells and provide mechanistical insight. 

Here, we first demonstrate that osteosarcoma-derived EVs can be rapidly internalized by lung fibroblasts in vitro in a dose- and time-dependent manner. Additionally, EV treatment lead to upregulation of αSMA stress fibers and caused the differentiation of fibroblasts into myofibroblast phenotypes associated with increased invasive capabilities. Furthermore, osteosarcoma-derived EVs’ treatment of fibroblasts also resulted in enhanced MMP2 and MMP9 activity as well as increased fibronectin expression. Our data are in line with findings in several cancer cell models, in which it is now well established that both the generation of biologically active ECM fragments via ECM degradation as well as fibronectin deposition provide critical biomechanical and biochemical cues that guide the homing and arrest of bone marrow derived cells and circulating tumor cells to PMNs thereby promoting metastasis [24,25,26]. Similar to findings in bladder cancer, we also observed an increase in the levels of FAP, a known CAF marker [27]. 

Unlike studies in breast cancer and melanoma models in which horizontal transfer of miRNAs cargo caused fibroblast differentiation [22,28], we identify osteosarcoma-EV associated TGFβ1 as the major driver of pulmonary fibroblast activation in vitro. Both the pharmacological inhibition of TGFβ downstream signaling in fibroblasts and CRISPR-Cas9 mediated gene deletion of TGFβ1 in osteosarcoma cells prevented osteosarcoma-derived EVs from activating lung fibroblasts in vitro. Our findings are consistent with those in prostate and bladder cancer revealing exosomal TGFβ to trigger fibroblast differentiation [19,27]. The TGFβ1 levels detected in 143-B EVs were comparable to levels in EVs isolated from other cancer cell lines [19]. TGFβ is widely accepted to have a dual role as a tumor suppressor or tumor promoter based on the tumor type and stage of disease [29]. However, in osteosarcoma, TGFβ seems to mainly have a pro-tumoral effect with high TGFβ expression positively correlating with advanced-grade tumors in patients [30,31]. Disruption of the TGFβ signaling pathway in osteosarcoma cells either through the overexpression of a natural TGFβ/Smad signaling inhibitor Smad7 or through the use of TGFβ inhibitors such as halofuginone prevented the expression of TGFβ target genes, including *RANKL*, *IL11*, *CXCR4*, *VEGF* and osteopontin, which slowed down primary tumor growth, reduced bone osteolysis and metastatic burden [32,33]. TGFβ signaling in stromal cells can be mediated through tumor-derived EVs in both a membrane associated form as well as through intravesicular cargo [27]. Membrane associated TGFβ is usually expressed in a latent form and is supported with heparan sulphate side chains that are fundamental for mediating TGFβ signaling [34]. High levels of membrane-associated TGFβ was found in osteosarcoma-cell-line-derived EVs as well as in the serum of osteosarcoma patients compared to control subjects [16]. Our study is in agreement with the above-mentioned findings, and we hypothesize that the delivery of EV-associated TGFβ to distant sites of metastasis may present a novel mechanism through which osteosarcoma primary tumors mediate PMN formation. 

Recently, it has been reported that osteosarcoma cell-derived EVs contain other growth factors/cytokines that promote angiogenesis in endothelial cells [35]. An activated endothelium along with neo-angiogenesis is characteristic of the PMN and is associated with immune/tumor cell infiltration via upregulated adhesion molecules [36,37]. To our knowledge, none of the previous studies have directly addressed if cancer-derived EVs alter the expression profile of adhesion molecules in endothelial cells. We, therefore, assessed the effect of EVs on endothelial cell activation. We observed a very modest difference in transcript levels of adhesion molecules upon osteosarcoma cell-derived EV stimulation, implicating that EVs do not play a direct role in endothelial cell activation in vitro. However, in the current study, we did not investigate potential indirect effects, which might include EV-regulated cytokine release from other stromal cells that can drive increased adhesion or endothelial permeability [38]. Although widely used as an experimental model in cancer biology research, our in vitro model employing immortalized established cancer cell lines has several drawbacks. Two-dimensional cell culture models do not mimic the natural tumor tissue architecture, including cell–cell and cell–matrix interactions, and are not representative of the complete diversity of tumors such as tumor and stromal cell heterogeneity [39]. Nevertheless, the in vitro approach used here was instrumental for dissection of the molecular mechanisms of EV-mediated intercellular communication. A broader biological significance of osteosarcoma-derived EVs is yet to be determined. Future studies should focus on establishing complex 3D in vitro systems, such as patient-derived organoids, or sophisticated mouse models that enable investigation of EV-mediated communication between osteosarcoma cells at the primary tumor site and lung stromal cells in a setting that more closely recapitulates the physiological environment.

Taken together, our findings provide novel insights into the functional relevance of EV-mediated communication between osteosarcoma cells and normal pulmonary fibroblasts. We deliver strong evidence that osteosarcoma cell-derived EVs can induce lung fibroblast reprogramming and thereby direct fibroblast activation and differentiation towards a myofibroblast/CAF phenotype through EV-associated TGFβ1 and SMAD2 pathway activation. In conclusion, we highlight osteosarcoma cell-derived EVs as important modulators of the distant microenvironment. Finally, a better understanding of EVs and their interaction with the cellular components of the pre-metastatic niche may lead to the onset of novel therapies that target stromal cell–tumor cell interactions and halt osteosarcoma metastatic progression. 

## 4. Materials and Methods

### 4.1. Cell Culture and Reagents

The human osteosarcoma cell lines SaOS-2 (HTB-85) and 143-B (CRL-8303) and the normal lung fibroblast MRC-5 (CCL-171) cells were obtained from ATCC (Gaithersburg, MD, USA). WI-38 (90020107) Lung fibroblast cells were purchased from Sigma-Aldrich (St. Louis, MO, USA). Human osteosarcoma HUO9 cells were kindly provided by M. Tani (National Cancer Center Hospital, Tokyo, Japan) and the murine osteosarcoma cell lines Dunn and LM8 by T. Ueda (Osaka National Hospital, Osaka, Japan). Osteosarcoma cells and lung fibroblasts were cultured in DMEM (4.5 g/L glucose)/HamF12 (1:1) medium (61965026 and 1765029, Thermo Fischer Scientific, Paisley, UK) or DMEM only (31966021, Thermo Fischer Scientific, Paisley, UK), respectively, and supplemented with 10% heat-inactivated fetal bovine serum (FBS; 10500, Thermo Fischer Scientific, Paisley, UK), referred to as complete medium hereafter. The cells were grown at 37 °C in a humidified atmosphere of 5% CO2 and 95% air. All cells were confirmed to be mycoplasma negative. The human cell lines were authenticated by short tandem repeat DNA profiling (Microsynth, Balgach, Switzerland) with a PowerPlex^®^16HS system (DC2101, Promega, Madison, WI, USA) and by comparison with the German Collection of Microorganisms and Cell Cultures database. Recombinant human transforming factor beta 1 (TGFβ1; 580702, Biolegend, San Diego, CA, USA), recombinant tumor necrosis factor α (TNFα; 300-01A, Peprotech, London, UK) and platelet-derived growth factor-B (PDGF-B; 100-14B, Peprotech, London, UK) were used at a concentration of 10 ng/mL, 10 ng/mL and 20 ng/mL, respectively. SB-431542 hydrate (S4317-5MG, Sigma Aldrich, St. Louis, MO, USA), a selective inhibitor of ALK5/TGFβ type I receptor, was used at a concentration of 10 μM.

### 4.2. EV Isolation, Fibercell Hollow-Fiber Bioreactor Culture and Bioreactor EV Purification

EVs were purified by differential centrifugation processes as described previously [40]. Briefly, osteosarcoma cells were seeded into 15 cm dishes in complete medium. After reaching 70–80% confluency, the cells were washed with PBS and subsequently incubated in serum free medium (SFM) for 48 h. The conditioned medium was first centrifuged at 600× *g* for 10 min to remove cells and then centrifuged at 2000× *g* for 15 min to remove debris. The resulting precleared supernatant was then ultracentrifuged in a 70Ti rotor (Beckman-Coulter, Brea, CA, USA) at 10,000× *g* at 4°C for 30 min and then sequentially ultracentrifuged at 100,000× *g* and 4 °C for 70 min. The supernatant was discarded and the pellet was washed in PBS using the same ultracentrifuge conditions. The pellet containing the purified EVs was collected in PBS and passed through a 0.2 μm pore filters. EVs isolated by this method are referred to as 2D EVs. 

Alternatively, to obtain large amounts of EVs from 143-B cells, a bioreactor culture was set-up. A medium-sized, hollow-fiber culture cartridge with a 20 kDa molecular weight cut-off (C2011, Fibercell Systems; Frederick, MD, USA) was pre-cultured according to the manufacturer’s instructions. 143-B cells stably expressing mCherry, luciferase and LacZ [41,42] were expanded in conventional culture flasks and 10^8^ cells were used to inoculate the hollow-fiber culture cartridge. Cells were then slowly adapted to bioreactor culture conditions over two weeks by gradually increasing the concentration of protein-free medium (1:1 DMEM/Ham’s F12 + 10% Fibercell Systems chemically defined medium for high density cell culture (CDM-HD; Fibercell Systems, Frederick, MD, USA) protein-free supplement + 100 U/mL penicillin/streptomycin). Bioreactor-conditioned medium (20 mL) was collected for each harvest 3–5 times per week depending on the glucose concentration of the medium reservoir. Harvests were precleared of cells and debris by sequential centrifugation at 300× *g* for 10 min and 2500× *g* for 15 min. The supernatant was ultracentrifuged in a 70Ti rotor (Beckman-Coulter, Brea, CA, USA) at 20,000× *g* for 45 min at 4 °C. The supernatant was passed through a 0.22 µm bottle top filter (431118, Corning^®^, Somerville, MA, USA) and then ultracentrifuged at 100,000× *g* for 70 min at 4 °C. Pellets were resuspended in PBS and pooled together. EVs were then purified on a 30% sucrose-tris-D2O cushion [40]. Purified EVs were resuspended in PBS and sterilized by passing through a 0.22 µm Steriflip^®^ filter (Merck-Millipore, Molsheim, France). The protein content in EV preparations was determined with a bicinchoninic acid (BCA) Protein Assay Kit (23225, Thermo Fisher Scientific, Paisley, UK). The EVs were either used immediately or stored at −80 °C until use in experimental procedures. For functional in vitro studies, the recipient cells were treated with a 20 µg/mL of EVs unless stated otherwise. EVs isolated by this method are referred to as 3D EVs.

### 4.3. Iodixanol Density Gradient Centrifugation

For a density-based purification of EVs, a discontinuous iodixanol gradient was prepared [43]. Solutions (40% (*w/v*), 20% (*w/v*), 10% (*w/v*) and 5% (*w/v*)) of iodixanol were made by diluting OptiPrepTM (60% (*w/v*) with 0.25 M sucrose/10 mM Tris, pH 7.5 (Stemcell Technologies, Vancouver, BC, Canada). Solutions were laid in a bottom-up approach in 12 successive layers of 1 mL each. Two-dimensional 143-B EVs were added at the top of the gradient and were ultracentrifuged at 20,000× *g* for 12 h at 4 °C. A series of 12 fractions of 1 mL each were then collected, and each fraction was diluted in 20 mL of PBS. Fractions were then ultracentrifuged at 100,000× *g* using a 70Ti rotor for 70 min at 4 °C. Pellets were lysed using radioimmunoprecipitation assay (RIPA) buffer and proteins were separated by SDS-PAGE and blotted for EV-associated markers.

### 4.4. Nanoparticle Tracking Analysis (NTA)

The size distribution and concentration of nanoparticles in 2D and 3D EV isolations were assessed using the NanoSight NS300 device (Malvern Panalytical, Malvern, UK). Samples were diluted between 1:100 and 1:1000 in order to obtain an optimal detectable concentration of 40–80 particles/frame. sCMOS camera levels were kept at 14–15 depending on the concentration of samples. Samples were injected in the 488 nm laser chamber with a constant output controlled by a syringe pump. Five 60 s video recordings were performed for each sample. NTA software (NTA 3.1 Build 3.1.54, Malvern Panalytical, Malvern, UK) was used to analyze the data with a detection threshold of 3. GraphPad Prism (v8.4.2, GraphPad Software, San Diego, CA, USA) was used to integrate the five technical measurements of each sample.

### 4.5. Flow Cytometry Analysis of EV-Coated Beads

The expression of exosomal markers CD63 and CD81 in osteosarcoma-derived EVs was determined by flow cytometry as described earlier [44]. Briefly, 2D EVs were coupled to 4 µm aldehyde-sulfate latex beads (Thermo Fisher Scientific; A37304, Paisley, UK) by mixing and rotation in a benchtop shaker at room temperature for 15 min. The mixture was diluted with 600 μL of PBS and mixed overnight at 4 °C. To stop the reaction, 400 µL of 1 M glycine was added and mixed at room temperature for 1 h. The EV-coated beads were centrifuged at 14,000× *g* for 1 min at room temperature and blocked with 10% BSA in PBS by rotation for 45 min at room temperature. After centrifugation at 14,000× *g* for 1 min at room temperature, EV-coated beads were then resuspended in 60 μL of 2% BSA in PBS, and split equally into three tubes and subsequently incubated with either secondary antibody control alone, anti-CD63 (556019, BD Pharmingen, San Jose, CA, USA) or anti-CD81 (10630D, Thermo Fisher Scientific, Paisley, UK) antibodies with rotation from 30 min at room temperature. After washing, the beads were mixed with donkey anti-mouse Alexa Fluor 546 secondary antibody (A10036, Thermo Fisher Scientific, Paisley, UK) at room temperature for 1 h. After washing twice with 2% BSA in PBS, EV-coated beads were analyzed for CD63 and CD81 using the LSRFortessa analyzer and FlowJo v10.5.3 software (BD Biosciences, Ashland, OR, USA). 

### 4.6. EV Labelling and Uptake

To study EV uptake by recipient cells, 2D EVs were prelabeled with a fluorescent lipophilic membrane dye using a PKH67 Cell Linker Kit (PKH67GL, Sigma-Aldrich, St. Louis, MO, USA) according to the manufacturer’s instructions with minor modifications. Briefly, 200 µL of Diluent C and 2 µL of PKH67 dye were mixed separately. EVs (100 µg) resuspended in 200 µL PBS were incubated with the PKH67 solution for 5 min at room temperature. The labelling reaction was quenched by adding 2 mL of 7.5% BSA in PBS. To remove the excess dye, samples were resuspended in PBS and ultracentrifuged in a 70Ti rotor (Beckman-Coulter, Brea, CA, USA) at 100,000× *g* and 4 °C for 70 min. The pellet was resuspended and washed again in PBS under the same ultracentrifugation conditions. The final pellet was resuspended in 100 µL of PBS. For uptake experiments, 2 × 10^5^ of WI-38 and MRC-5 cells were seeded per well in 12-well plates. On the next day, the cells were washed with PBS and 2 µg or 10 µg of labelled EVs in DMEM medium containing 1% Penicillin/Streptomycin (P0781, Sigma-Aldrich, St. Louis, MO, USA) was then added. The cells were cultured at 37 °C or 4 °C for the indicated time period. To inhibit cellular endocytosis, the cells were pretreated with 5 µM of Dyngo-4a (SML0340, Sigma-Aldrich, St. Louis, MO, USA) for 30 min prior to EV incubation and left present throughout the experiment. After incubation with the EVs, cells were washed and trypsinized. After washing with PBS, the cells were analyzed using the LSRFortessa and FlowJo v10.5.3 software (BD Biosciences, Ashland, OR, USA). The experiment was performed with 3 biological replicates and 20,000 events were analyzed per condition. The data are represented both with the mean fluorescence intensity (MFI) as well as the % of positive cells. 

### 4.7. Western Blotting and Zymograhy Analysis

Protein extraction and Western blotting were performed as previously described [45]. Primary antibodies used were anti-αSMA (ab5694, Abcam, Cambridge, UK; dilution 1:5000), anti-FAP (PA5-51057, Thermo Fisher Scientific; dilution 1:1000), anti-Tubulin (sc-23948, Santa Cruz Biotechnology, Dallas, TX, USA; dilution 1:5000), anti-GAPDH (G8795, Sigma-Aldrich, St. Louis, MO, USA; dilution 1:5000), anti-CD9 (13174S, Cell Signaling Technology, Denver, MA, USA; dilution 1:1000), anti-Fibronectin (F3648, Sigma-Aldrich, St. Louis, MO, USA; dilution 1:1000), anti-TGFβ (GTX130023-S, GeneTex, Irvine, CA, USA; dilution 1:1000), anti-Phospho-Smad2 (Ser465/467) (138D4, Cell Signaling Technology; dilution 1:1000) and anti-Smad2/3 (D7G7, Cell Signaling Technology; dilution 1:1000). Anti-mouse HRP-conjugated secondary antibodies were obtained from Thermo Fisher Scientific (A10668, Paisley, UK; dilution 1:3000). Anti-rabbit HRP-conjugated secondary antibodies were purchased from Santa Cruz Biotechnologies (sc-2054, Dallas, TX, USA; dilution 1:3000). Peroxidase activity was visualized with Immobilon chemiluminescence substrate (WBKLS0500, Merck-Millipore, Molsheim, France) and a ChemiDoc XRS+ Imaging System (Bio-Rad, Hercules, CA, USA). The detection and quantitation of bands was performed using Image Lab v6.1.0 (Bio-Rad, Hercules, CA, USA). 

The activity of metalloproteinases MMP2 and MMP9 was analyzed using gelatinolytic zymography as described previously [46]. Briefly, a 2× Novex™ Sample Buffer (LC2673, Thermo Fischer Scientific, Paisley, UK) was added to MRC-5 lung fibroblasts conditioned medium treated with/without 2D 143-B EVs (20 µg/mL) for 48 h and run through a 10% acrylamide gel containing 0.1% gelatin. Samples were washed with Novex™ Zymogram Renaturing Buffer (LC2670, Thermo Fischer Scientific, Paisley, UK) for 15 min and incubated with Novex™ Zymogram Developing Buffer (LC2671, Thermo Fischer Scientific, Paisley, UK) for 17 h at 37 °C. Samples were stained with Coomassie blue for 1 h at room temperature and destained with 4% methanol in 10% acetic acid. MMP activity was visualized with a ChemiDoc™ MP Imaging System (Bio-Rad, Hercules, CA, USA).

### 4.8. Negative Staining Transmission Electron Microscopy

A total of 10 µL of either the 2D or 3D EV suspension was pipetted on a glow-discharged, carbon/formvar-coated transmission electron microscopy grid and incubated for 20 min, followed by rinsing on a droplet of 100 µL PBS. Subsequently, the specimen was fixed with 1% glutaraldehyde in H2O, rinsed twice with H_2_O, and incubated with 1% of uranyl acetate in H_2_O; all of these steps were performed for 5 min and at room temperature. Thereafter, the samples were incubated with a solution of 0.3% uranyl acetate (Polysciences, Hirschberg an der Bergstraße, Germany) and 1.8% methylcellulose (M0512, Sigma-Aldrich, St. Louis, MO, USA) in H_2_O for 10 min on ice prior to removing the solution with a filter paper and air drying. Finally, all samples were imaged in a CM100 transmission electron microscope (FEI-Thermo Fisher Scientific, Hillsboro, OR, USA) at an acceleration voltage of 80 keV using an Orius 1000 digital camera (Gatan, Munich, Germany).

### 4.9. RNA Isolation, Reverse Transcription and qRT-PCR

Total RNA from cells treated with either 2D or 3D EVs for 24 h or control untreated cells was extracted using a RNeasy Mini Kit (74194, Qiagen, Hilden, Germany), accompanied by on-column DNase-treatment (79254, Qiagen, Hilden, Germany) according to the manufacturer’s protocol. Total RNA (0.5–1 µg) was transcribed to cDNA with a High-Capacity cDNA Reverse Transcription Kit with RNAse Inhibitor (4374966, Applied Biosystems, Carlsbad, CA, USA) according to the manufacturer’s instructions. The cDNA was diluted in nuclease free water and real-time qPCR was conducted on cDNA equivalent to 10 ng of starting RNA with a Power SYBR Green PCR Master Mix (4367659, Applied Biosystems, Carlsbad, CA, USA) on a StepOne-Plus Real-Time PCR System (Applied Biosystems, Carlsbad, CA, USA). The denaturation was performed for 10 min at 95 °C followed by 40 PCR cycles for 15 s at 95 °C and for 1 min at 60 °C. The reactions were run in technical triplicates. The analysis was done with StepOne Software version 2.1 (Applied Biosystems, Carlsbad, CA, USA). Relative expression levels were calculated by using the 2(-ΔCT) method, normalized to GAPDH. The primers used are detailed in Appendix A. The experiments were repeated at least three times.

### 4.10. Adhesion Assay

The activation of endothelial cells by osteosarcoma-derived 2D EVs was investigated using an endothelial-tumor cell adhesion assay as previously described [47]. Human umbilical vein endothelial cells (HUVECs) were purchased from ATCC (CRL-1730™) and were cultured in EGM-2 Bullet kitTM (CC-3162, Lonza, Basel, Switzerland). Briefly 35,000 HUVECs per well were seeded in 96-well plates in EGM-2 Bullet kitTM (CC-3162, Lonza) 36 h before the adhesion assay. Following overnight incubation, HUVECs were treated with fresh EBM-2 basal medium (00190860, Lonza) supplemented with PBS, 20 µg/mL EVs or TNFα (300-01A, Peprotech, London, UK) for 12 h. 143-B cells were stained with CellTracker™ Orange CMTMR Dye (C2927, Thermo Fischer Scientific, Paisley, UK) according to the manufacturer’s instructions. 143-B cells (104) were then allowed to adhere to HUVECs at 37 °C for 15 min. Non-adherent cells were then washed out twice with PBS and cells were fixed with 4% formalin at RT for 10 min. Images of adherent fluorescent cells were captured using a Zeiss Observer Z1 microscope (Zeiss, Munich, Germany) and quantified using ImageJ 1.52n. 143-B cell adhesion to HUVECs was assessed in three independent experiments in triplicate wells for each treatment condition.

### 4.11. Spheroid Invasion Assay

The invasive properties of WI-38 and MRC-5 fibroblasts were investigated using a spheroid invasion assay performed and quantified as previously described [48]. Briefly, 2500 cells per well were seeded in 100 µl of DMEM containing 10% FBS into a cell-repellent 96-well microplate (650790, Greiner Bio-one, Frickenhausen, Germany). After overnight incubation at 37 °C, the cells formed spheroids, 70 μL of medium was removed from each well and the remaining medium with spheroid was overlaid with 2.5% bovine collagen I (5005, Advanced BioMatrix, Carlsbad, CA, USA). Following the polymerization of collagen, fresh serum-free DMEM supplemented with PBS, 20 µg/mL 2D EVs or 20 ng/mL of PDGF-B (100-14B, Peprotech, London, UK) was added. The cells were allowed to invade the collagen matrix for 48 h, after which they were stained with Hoechst (62249, Thermo Fischer Scientific, Waltham, MA, USA). Images were acquired on an Axio Observer 2 fluorescence microscope (Zeiss) using a 5× objective. Cell invasion is determined as the average of the distance invaded by the cells from the center of the spheroid as determined by the cell dissemination counter software aSDIcs. Three to five spheroids per condition were analyzed and three independent experiments were performed.

### 4.12. Generation of CRISPR/Cas9-Mediated Knockout Cells

Single-guide RNAs (sgRNAs) against *TGFB1* gene were designed with the CRISPRdirect online tool (http://crispr.dbcls.jp; [49]). Only highly specific target sites were selected and the respective sequences are listed in Appendix A. A non-targeting control sgRNA was chosen from the study of Morgens et al. [50]. Oligonucleotides specific for the target sites were synthesized with BsmBI restriction site overhangs by Microsynth (Balgach, Switzerland) and then annealed and cloned into the lentiCRISPRv2 transfer plasmid, a gift from Feng Zhang (Addgene plasmid # 52961; [51]), following the provided protocol. Lentiviral particles were produced by the co-transfection of the lentiCRISPRv2 plasmid, containing the respective gRNA-sequence, with the packaging plasmids pCMV-VSV-G (a gift from Bob Weinberg; Addgene plasmid # 8454) and psPAX2 (a gift from Didier Trono; Addgene plasmid # 12260) into HEK-293T cells using LipofectamineTM 3000 (L3000008, Thermo Fisher Scientific, Paisley, UK) according to the manufacturer’s instructions. For transduction, human osteosarcoma cells were incubated for 24 h with supernatant containing the viral particles and 8 µg/mL polybrene (TR-1003-G, Merck Millipore, Molsheim, France). Subsequently, the cells were selected with 5 µg/mL puromycin (A1113803, Thermo Fisher Scientific, Paisley, UK) for 7 days. The cell lines are designated as control 143-B NT and knockout cells 143-B KO#1 and 143-B KO#2. The efficiency of the knockouts was tested with Western blotting.

### 4.13. ELISA

The Quantikine^®^ ELISA human TGFβ1 Immunoassay (DB100B, R&D systems^®^, Minneapolis MN, USA) was used to assess EV-associated TGFβ1 from 2D EVs isolated from 143-B, SaOS-2 and Hu09 cells. Briefly, 10 µg of EVs was used from independently isolated extracts. The assay was processed and analyzed according to the manufacturer’s instructions.

### 4.14. Statistical Analysis

Statistical analyses were performed using GraphPad Prism 8 (v8.4.2, GraphPad Software, San Diego, CA, USA). The statistical significance of differences between groups was determined using Student’s *t*-test (unpaired, two-tailed) or a one-way ANOVA test with a Bonferroni post-hoc test. The experiments were performed in at least three independent biological replicates, and the results are shown as the mean ± standard error of the mean (SEM). The results were considered significant when *p* < 0.05.

## Figures and Tables

**Figure 1 ijms-21-05451-f001:**
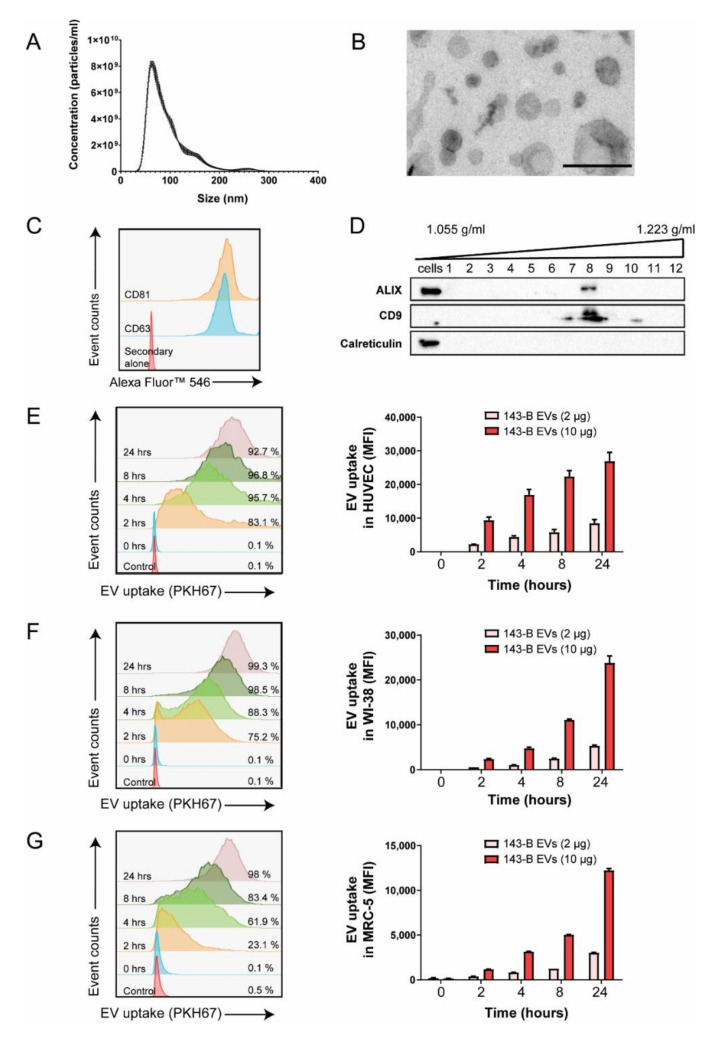
Characterization of osteosarcoma 143-B-derived extracellular vesicles (EVs) and their uptake by recipient cells. (**A**) 143-B EV size distribution assessed by NanoSight analysis. (**B**) Transmission electron micrographs of 143-B EVs. Scale bar, 250 nm. (**C**) Representative histogram of flow cytometry analysis of exosomal markers CD63 and CD81 in 143-B EV-coated beads. (**D**) Western blot analysis of EV markers (ALIX, CD9) and a cellular marker (calreticulin) in 143-B whole cell protein extracts and in eluted EV fractions from iodixanol based density gradient ultracentrifugation of 143-B EVs. Internalization of PKH67-labelled 143-B EVs by human umbilical vein endothelial cells (HUVEC) (**E**), WI-38 (**F**) and MRC-5 (**G**) cells, as measured by flow cytometry. Representative histograms indicating the percentage of cells internalizing EVs (10 µg) over time (left panels) and the quantification of mean fluorescence intensity (MFI) indicating dose- and time-dependent EV uptake (right panels). The data represent means ± SEM from three independent experiments.

**Figure 2 ijms-21-05451-f002:**
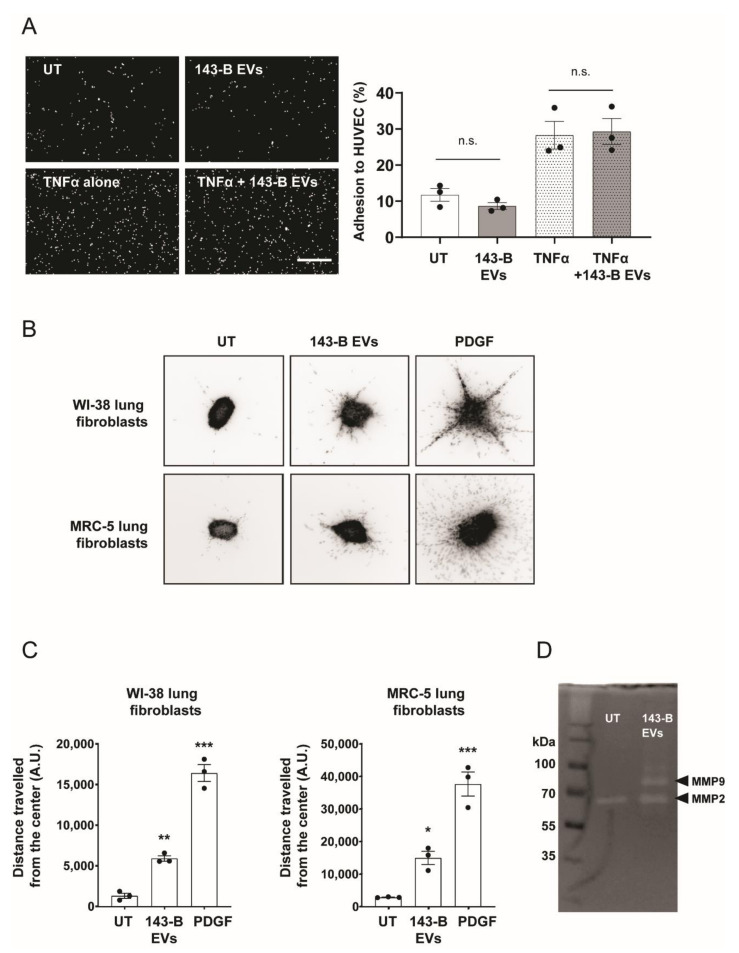
Functional consequences of 143-B EV treatment of endothelial cells and lung fibroblasts. (**A**) Representative images (left panel) and quantification (right panel) of 143-B cell adhesion to HUVEC monolayers. Fluorescently labelled tumor cells were allowed to adhere for 15 min to HUVEC cells left untreated (UT) or treated for 12 h with 143-B-derived EVs (20 μg/mL) in the presence or absence of TNFα (10 ng/mL). Scale bar, 500 µm. n.s., not significant. (**B**) Representative images of the spheroid invasion assay with WI-38 and MRC-5 fibroblasts after 48 h of incubation with 143-B EVs (20 μg/mL) or platelet-derived growth factor (PDGF) (20 ng/mL) and (**C**) the quantification of distances from spheroids centers using cell dissemination counter software aSDIcs (right panels); (* *p* < 0.05, ** *p* < 0.01, *** *p* < 0.001, one-way ANOVA with Bonferonni’s post hoc test). **(D**) Zymography analysis of MMP-2 and MMP-9 activity in untreated MRC-5 fibroblasts or treated with 143-B-derived EVs (20 μg/mL) for 48 h. Values in (**A**) and (**C**) represent the means ± SEM from three independent experiments.

**Figure 3 ijms-21-05451-f003:**
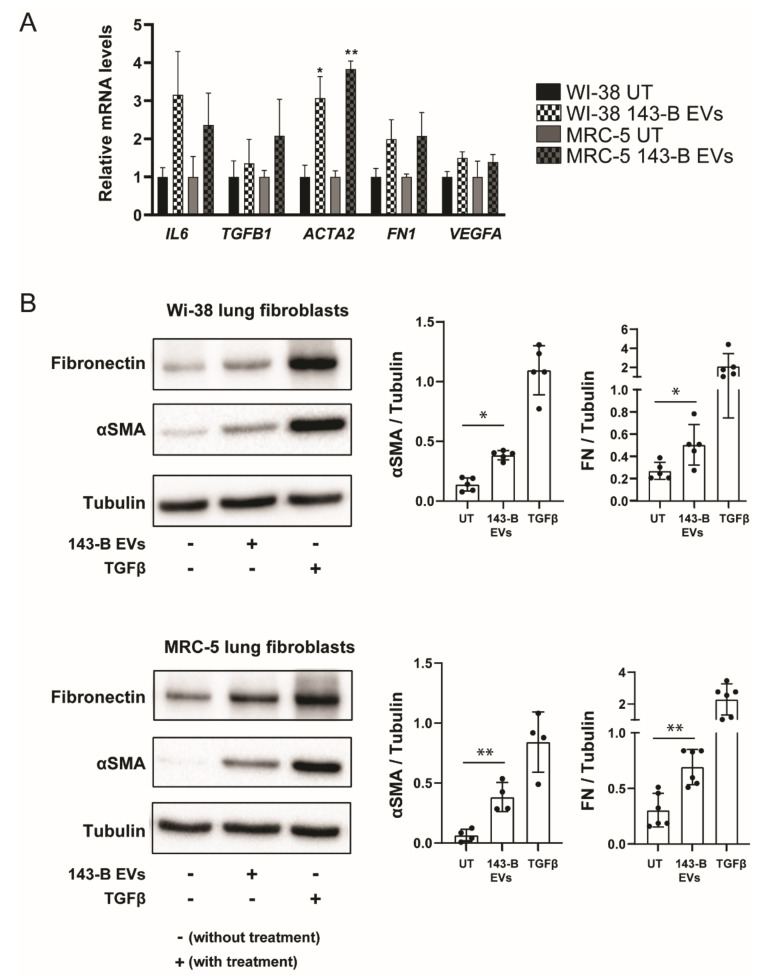
143-B EVs induce lung fibroblast differentiation in vitro. (**A**) Relative mRNA expression levels of indicated genes in lung fibroblasts following incubation with 143-B-derived EVs (20 μg/mL) for 24 h. Transcript levels are normalized to glyceraldehyde 3-phosphate dehydrogenase (GAPDH) and expressed as fold change relative to the untreated control (UT). At least three independent experiments were performed (* *p* < 0.05, unpaired Student’s t-test). (**B**) Fibronectin and α-smooth muscle actin (αSMA) protein levels in whole cell extracts of lung fibroblasts treated with 143-B-derived EVs (20 μg/mL) or soluble transforming growth factor β1 (TGFβ1) for 48 h were examined by Western blot analysis. Tubulin was used as a protein loading control. Representative Western blots (left panels) and respective quantitative analysis of at least three independent experiments (right panels) (* *p* < 0.05, * *p* < 0.01, unpaired Student’s *t*-test) are shown.

**Figure 4 ijms-21-05451-f004:**
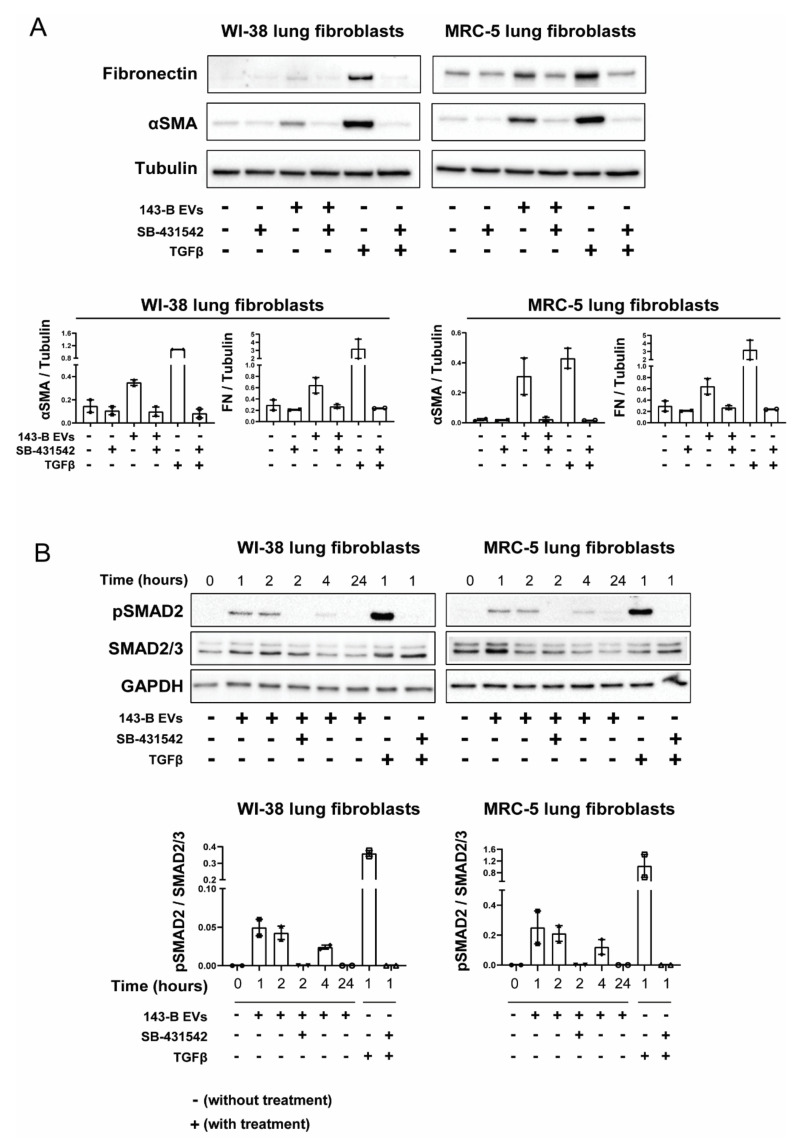
Osteosarcoma cell-derived EVs trigger αSMA expression in lung fibroblasts through TGFβ. (**A**) Western blot analysis of fibronectin and αSMA protein expression in WI-38 and MRC-5 fibroblasts upon exposure to 20 μg/mL 143-B-derived EVs and 10 ng/mL of soluble TGFβ1 for 48 h with or without pre-treatment with a TGFβ receptor 1 inhibitor SB-431542 (10 μM) for 30 min. (**B)** Effects of 143-B EVs on SMAD2 phosphorylation in lung fibroblasts. Serum-starved (24 h) WI-38 and MRC-5 cells were left untreated or were pre-treated for 30 min with SB-431542 (10 μM) prior to incubation with 143-B EVs (20 μg/mL) or TGFβ1 (10 ng/mL) for the indicated periods of time. Protein expression of phosphorylated SMAD2, total SMAD2/3 and GAPDH was evaluated by Western blot. Representative Western blots (upper panels) and respective quantitative analysis of two independent experiments (lower panels) are shown.

**Figure 5 ijms-21-05451-f005:**
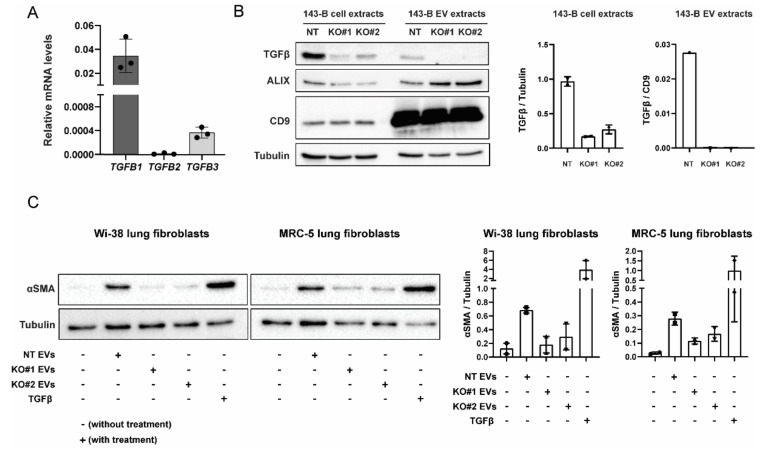
CRISPR-Cas9-mediated knockout of *TGFB1* prevents 143-B-derived EVs from inducing lung fibroblast differentiation. (**A**) mRNA expression levels of indicated *TGFβ* isoforms (relative to *GAPDH*) in 143-B osteosarcoma cells as determined by qRT-PCR. (**B**) Western blot analysis of TGFβ1 protein expression was used to validate the efficiency of CRISPR-Cas9-mediated genome editing in non-targeting control (NT) 143-B, TGFβ1 KO#1 143-B, and TGFβ1 KO#2 143-B cell extracts and, respectively, derived-EV protein extracts. Cell extracts were harvested 4-5 passages after the initial viral transduction. Protein levels of EV markers (CD9 and ALIX) were examined by Western blot. (**C**) Western blot analysis of αSMA expression in WI-38 fibroblasts and MRC-5 fibroblasts following 48 h incubation with 20 μg/mL of EVs derived from control NT 143-B cells, TGFβ1 KO#1 143-B cells and TGFβ1 KO#2 143-B cells. Soluble TGFβ (10 ng/mL was used as the positive control. Tubulin was use as a loading control. Representative Western blots (left panels) and respective quantitative analysis of two independent experiments (right panels).

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
