# Peer review of "Osteosarcoma-Derived Extracellular Vesicles Induce Lung Fibroblast Reprogramming"

_ijms, 2020, doi:10.3390/ijms21155451_

Round 1

Reviewer 1 Report

The manuscript by Mazumdar et al. entitled “Osteosarcoma-derived Extracellular Vesicles Induce Lung Fibroblast Reprogramming” aims to characterize a tumor-supportive role for metastatic osteosarcoma-derived EVs in inducing lung fibroblast reprogramming. In particular, the authors show that EVs secreted from osteosarcoma cells are internalized by endothelial cells and lung fibroblasts in vitro and induce phenotypic changes in lung fibroblasts including elevated expression of gene products reflective of fibroblast activation and myofibroblast differentiation. Finally, the authors demonstrate that EV-associated TGF-β secreted from osteosarcoma cells in part drives these phenotypic changes by activation of SMAD signaling in lung fibroblasts. Overall the manuscript is clear and well-written, however several points should be addressed before publication.

Major points:

  1. Several techniques used to purify EVs are described in the methods, including differential centrifugations (with variations in speed and time of spins across analogous steps), filtration, sucrose cushion purification, and density gradient purification. The authors should clarify which techniques were used for each experiment.
  2. The authors should comment on why EVs measured by NTA range from 30-300 nm (line 86), after final EV pellets were passed through a 0.2-micron filter.
  3. PKH67 dye is well known to label co-precipitating lipoparticles or proteins in EV preparations, and may form dye micelles independent of vesicles or co-contaminants present in solution. Subsequent purification by density gradient or sucrose cushion is essential to remove dye aggregates prior to recipient cell uptake analyses. Ultracentrifugation wash alone has been demonstrated to be insufficient to remove these dye micelles (van der Vlist et al., Nature Protocols, 2012, PMID: 22722367).
  4. How were EV quantities for transfer experiments determined? Does the quantity of EVs added to recipient cells approximate physiologic conditions? One way the authors could begin to address this question in vitro is via transwell co-culture of 143-B cells and lung fibroblasts to show a similar phenotype. Without relevant in vivo models, the authors should discuss the limitations of drawing biologic conclusions of EV function from cell culture models.
  5. Figure 3B: Levels of fibronectin and αSMA should be quantitated across replicate experiments for statistical comparison.
  6. Figure 4: Quantitation of western blot signals from all experimental replicates should similarly be performed. How do levels of TGF-β in added EVs compare to levels of soluble TGF-β used to treat cells?
  7. Figure 5B: Additional EV biomarkers should be examined in EV extracts from CRISPR cells.

Minor point:

  1. Line 358: Sucrose density gradient centrifugation methods: it appears as if iodixanol (Optiprep) was used, not sucrose. This should also be corrected in the text (e.g. line 93) if appropriate.

Reviewer 2 Report

In the present study, the Authors evaluated the potential of osteosarcoma cell-derived EVs to drive myofibroblast/cancer-associated fibroblast differentiation. In particular, they demonstrated that TGFβ1 in osteosarcoma cell-derived EVs is responsible for lung fibroblast differentiation. The matter is intriguing; however, the study has some weakness as detailed below.

Several studies have demonstrated that αSMA is also not truly specific for CAFs, as further cell types including smooth muscle cells and pericytes express significant levels of the protein, therefore not recommending the use of αSMA alone as a primary marker for CAF identification. Moreover, increased FN expression levels were ascertained also in normal fibroblasts exposed to various stimuli, including tumor cells.  Hence, in addition to αSMA and FN, the Authors should evaluate in WI-38 and MRC-5 cells upon 143-B EV treatment the expression of a typical CAF marker namely fibroblast activation protein alpha (FAP) (preferably not as mRNA levels) in order to verify the EV capability to induce fibroblast differentiation/activation into CAFs.

As the Authors state that TGFβ is responsible for fibroblast-activating effect of 143-B EVs (as demonstrated by using SB-431542 as well as CRISPR-Cas9 technology), they should ascertain whether EVs isolated from poorly metastatic osteosarcoma cells, which did not increase αSMA expression in lung fibroblasts, do not secrete TGF-β1 in comparison with EVs isolated from osteosarcoma cells.

Author Response

Response to Reviewer 2 Comments

Point 1: In the present study, the Authors evaluated the potential of osteosarcoma cell-derived EVs to drive myofibroblast/cancer-associated fibroblast differentiation. In particular, they demonstrated that TGFβ1 in osteosarcoma cell-derived EVs is responsible for lung fibroblast differentiation. The matter is intriguing; however, the study has some weakness as detailed below.

Several studies have demonstrated that αSMA is also not truly specific for CAFs, as further cell types including smooth muscle cells and pericytes express significant levels of the protein, therefore not recommending the use of αSMA alone as a primary marker for CAF identification. Moreover, increased FN expression levels were ascertained also in normal fibroblasts exposed to various stimuli, including tumor cells.  Hence, in addition to αSMA and FN, the Authors should evaluate in WI-38 and MRC-5 cells upon 143-B EV treatment the expression of a typical CAF marker namely fibroblast activation protein alpha (FAP) (preferably not as mRNA levels) in order to verify the EV capability to induce fibroblast differentiation/activation into CAFs.

Response 1: We appreciate this very important point raised by the Reviewer. Following Reviewer's recommendation, we investigated the FAP protein expression levels in WI-38 and MRC-5 fibroblasts upon 143-B EV treatment. This data has now been included in the revised manuscript (line 166, Supplementary Figure S4A).

Point 2: As the Authors state that TGFβ is responsible for fibroblast-activating effect of 143-B EVs (as demonstrated by using SB-431542 as well as CRISPR-Cas9 technology), they should ascertain whether EVs isolated from poorly metastatic osteosarcoma cells, which did not increase αSMA expression in lung fibroblasts, do not secrete TGF-β1 in comparison with EVs isolated from osteosarcoma cells.

Response 2:

We thank the reviewer for bringing into discussion an important point. As suggested by the Reviewer, we investigated TGF-β1 levels in EVs derived from human osteosarcoma cell lines with different metastatic potential by ELISA (line 226, 529, Supplementary Figure S6). Our data indicate that 143-B cell-derived EVs contain higher TGF-β1 levels in comparison to EVs isolated from poorly metastatic SaOS-2 and Hu09 cells. The TGF-β1 levels detected in 143-B EVs were comparable to levels in EVs isolated from other cancer cell lines [1]. This data is now included in Supplementary Figure S6 (line 226).

References:

  1. Webber, J.; Steadman, R.; Mason, M. D.; Tabi, Z.; Clayton, A., Cancer exosomes trigger fibroblast to myofibroblast differentiation. Cancer Res 2010, 70, (23), 9621-30.